# Transcriptional Profiling Reveals Mesenchymal Subtypes of Small Cell Lung Cancer with Activation of the Epithelial-to-Mesenchymal Transition and Worse Clinical Outcomes

**DOI:** 10.3390/cancers14225600

**Published:** 2022-11-15

**Authors:** Hae Jin Cho, Soon Auck Hong, Daeun Ryu, Sook-Hee Hong, Tae-Min Kim

**Affiliations:** 1Department of Medical Informatics, College of Medicine, The Catholic University of Korea, Seoul 06591, Republic of Korea; 2Cancer Research Institute, College of Medicine, The Catholic University of Korea, Seoul 06591, Republic of Korea; 3Department of Biomedicine & Health Sciences, College of Medicine, The Catholic University of Korea, Seoul 06591, Republic of Korea; 4Department of Pathology, College of Medicine, Chung-Ang University, Seoul 06974, Republic of Korea; 5Department of Internal Medicine, College of Medicine, The Catholic University of Korea, Seoul 06591, Republic of Korea

**Keywords:** small cell lung cancers, molecular taxonomy, mesenchymal tumors, epithelial-mesenchymal transformation, developmental trajectories

## Abstract

**Simple Summary:**

The aim of this study is to discover and characterize novel small cell lung cancer (SCLC) subtypes and further identify their relationship with existing SCLC subtypes. We identified SCLC-M (mesenchymal) tumors as SCLC subtypes that are distinctive of SCLC-I (inflamed) tumors. SCLC-M tumors showed elevated epithelial-to-mesenchymal transformation (EMT) activity but a low level of anticancer immune activity also with unfavorable clinical outcomes. Gene expression- and immunohistochemistry-based prediction suggests that SCLC-M tumors comprise approximately 5% of primary SCLC tumors. Given these unique molecular and clinical features, SCLC-M tumors should be taken into account in the clinical settings of SCLC management.

**Abstract:**

While molecular subtypes of small cell lung cancers (SCLC) based on neuroendocrine (NE) and non-NE transcriptional regulators have been established, the association between these molecular subtypes and recently recognized SCLC-inflamed (SCLC-I) tumors is less understood. In this study, we used gene expression profiles of SCLC primary tumors and cell lines to discover and characterize SCLC-M (mesenchymal) tumors distinct from SCLC-I tumors for molecular features, clinical outcomes, and cross-species developmental trajectories. SCLC-M tumors show elevated epithelial-to-mesenchymal transformation (EMT) and YAP1 activity but a low level of anticancer immune activity and worse clinical outcomes than SCLC-I tumors. The prevalence of SCLC-M tumors was 3.2–7.4% in primary SCLC cohorts, which was further confirmed by immunohistochemistry in an independent cohort. Deconvoluted gene expression of tumor epithelial cells showed that EMT and increased immune function are tumor-intrinsic characteristics of SCLC-M and SCLC-I subtypes, respectively. Cross-species analysis revealed that human primary SCLC tumors recapitulate the NE-to-non-NE progression murine model providing insight into the developmental relationships among SCLC subtypes, e.g., early NE (SCLC-A and -N)- vs. late non-NE tumors (SCLC-M and -P). Newly identified SCLC-M tumors are biologically and clinically distinct from SCLC-I tumors which should be taken into account for the diagnosis and treatment of the disease.

## 1. Introduction

Small cell lung cancers (SCLCs) are clinically and histologically distinct lung cancers [1]. SCLCs represent only 13% of all newly diagnosed cases of lung cancers but are particularly aggressive with dismal clinical outcomes [2]. SCLCs are frequently associated with early metastases (i.e., two thirds of patients at initial diagnosis have distant metastases) with limited treatment options [3]. Although initially responsive to cytotoxic therapies, these tumors eventually acquire resistance to conventional treatments with a median survival rate of under 2 years. In contrast to lung adenocarcinomas that have benefited from advancements in new anticancer strategies including targeted therapy and immunotherapy [4], markers that define the eligibility of SCLC to these treatment options remain largely unknown.

Genomic and transcriptomic analyses of SCLCs have provided a comprehensive molecular view of SCLC genomes and transcriptomes [5,6,7]. In addition to the near-universal *TP53* and *RB1* mutations observed in SCLC genomes [8], transcriptomic analyses have prompted the development of a ‘molecular taxonomy’ or molecular subtypes of SCLCs defined by lineage-specific transcription factors and regulators. From this perspective, SCLC tumors are not considered a single disease entity but instead constitute a diverse molecular spectrum of tumors with potential subtype-specific therapeutic vulnerabilities (reviewed in ref. [9]). The initial dichotomy between the two molecular subtypes of neuroendocrine (NE) SCLCs followed from the identification of the NE transcription factors *ASCL1* (achaete-scute homologue 1) and *NEUROD1* (neurogenic differentiation factor) as ASCL1-high and NeuroD1-high SCLC tumors, respectively [10,11]. Transcription factors or regulators defining subtypes of non-NE SCLCs lacking *ASCL1* and *NEUROD1* expression were also identified: *POU2F3* (POU class2 homeobox 3) [12] and *YAP1* (yes-associated protein 1) [13]. Overall, these transcriptional factors and regulators have been used to define four consensus molecular subtypes of SCLC: SCLC-A, -N, -P, and -Y with high levels of *ASCL1*, *NEUROD1*, *POU2F3*, and *YAP1* expression, respectively.

While it is unclear whether there are additional SCLC subtypes beyond these four SCLC subtypes, a subset of SCLCs with elevated immune activity have recently been identified and referred to as SCLC-I (inflamed) tumors [14]. SCLC-I tumors showed elevated transcriptional levels of immune genes, suggestive the presence of infiltrating immune cells, but also low expression of the SCLC subtype markers of *ASCL1*, *NEUROD1*, and *POU2F3*. One study reported favorable clinical outcomes for SCLC-I tumors and suggested that these tumors may benefit from immune checkpoint inhibitors combined with chemotherapy [14]. It is worth noting that SCLC-I subtypes are based on tumor-extrinsic properties such as the level of tumor-infiltrating lymphocytes rather than tumor-intrinsic properties such as the activity of transcription regulators. Mesenchymal tumors with high levels of tumor-infiltrating stromal cells and elevated expression of genes associated with the epithelial-to-mesenchymal transition (EMT) have been observed in diverse tumor types [15,16]. While the presence of mesenchymal SCLC tumors with resistance to conventional therapies has been previously proposed [17], it is not clear whether mesenchymal SCLC tumors represent a distinct SCLC subclass with respect to the current SCLC subtyping system.

Yes-associated protein 1 (*YAP1*) and transcriptional coactivator with PDZ-binding motif (*TAZ*) are downstream effectors of the Hippo pathway mediating development, cellular homeostasis, and disease [18]. Previous studies have revealed that *YAP1* and *TAZ* (also known as *WWTR1*) cooperate via the TEA domains (TEAD1–TEAD4) to transactivate genes regulating EMT, cell proliferation, and migration [19]. *YAP1* also responds to extracellular matrix stiffness and mechanical signals that promote the motility of cancer cells [20].

In this study, we analyzed publicly available SCLC transcriptome data to investigate the presence of mesenchymal SCLC tumors and to further characterize their molecular and clinical features. By applying unsupervised clustering, we identified six molecular subtypes among SCLC cohorts. A previous study reported that SCLC-I tumors showed both elevated EMT activity and antitumor immune activity [14]. We were able to distinguish mesenchymal SCLC (SCLC-M) tumors from SCLC-I tumors and SCLC-M tumors showed elevated expression of EMT-associated genes and low immune signature scores as well as worse clinical outcomes in contrast to SCLC-I tumors. SCLC-M tumors were consistently observed across cohorts of primary SCLC tumors and cell lines. We also investigated their associated transcriptional regulators, enriched molecular functions and signaling pathways, clinical outcomes, and developmental relationship with other SCLC subtypes using cross-species analysis.

## 2. Materials and Methods

### 2.1. Datasets

Three mRNA expression profiles including two of primary SCLC tumors and one of SCLC cell lines were obtained from public resources. Two primary SCLC RNA-seq datasets [5,7] were obtained from the European Genome-Phenome Archive (EGA) database (accession numbers EGAS00001000925 (*n* = 81) and EGAS00001004888 (*n* = 31), respectively). RNA-seq data of 50 SCLC cell lines were also obtained from the Cancer Cell Line Encyclopedia (CCLE) database (https://sites.broadinstitute.org/ccle/datasets) (accessed on 10 September 2022) [21]. Fastq files of RNA-seq data were uniformly processed; sequencing reads were aligned to the reference genome (hg19) using the splice-aware STAR aligner [22] and gene-level read counts were produced using HTseq [23]. Read counts were further converted into transcripts per million (TPM) levels and log2 transformed.

### 2.2. SCLC Subtyping

Primary SCLC RNA-seq data were subjected to non-negative matrix factorization (NMF) for unsupervised tumor subtyping [24]. To determine the appropriate number of clusters (*K* values), cophenetic correlation coefficients were calculated for a range of *K* values (2 to 10) and 1000 subsamples were tested for each *K* value. For functional annotation of SCLC subtypes, we employed gene set enrichment analysis (GSEA). Single sample GSEA (ssGSEA) was carried out with the GSVA R package using the hallmark gene set available in the MSigDB database (http://www.gsea-msigdb.org/gsea/msigdb/) (accessed on 12 September 2022) [25]. To determine major transcriptional regulators associated with SCLC subtypes, ARACNe (algorithm for the reconstruction of accurate cellular networks) was used [26]. We further used the VIPER (virtual inference of protein activity by enriched regulon analysis) algorithm [27] to estimate the extent of enrichment of SCLC subtype-specific genes with inferred transcription targets by ARANCe. We performed log-rank tests to estimate the significance of differences in overall survival among SCLC subtypes along with Kaplan–Meier survival curves. For SCLC subtyping of tumors in the extended datasets, we employed prediction analysis of microarrays (PAM) as available in the pamr R package [28]. Six SCLC subtypes identified in the datasets of George et al. [5] were used as training data to predict SCLC subtypes in the datasets of Rudin et al. [7] and CCLE [21]. The six SCLC subtypes for the George et al. [5] dataset were further compared with those made by different classification systems: the four SCLC subtype classification system (SCLC-A/-N/-P/-I) and that based on four transcription factors or regulators (SCLC-A/-N/-P/-Y) [9,14]. Relationships among clusters across the subtyping systems were visualized using Caleydo [29].

### 2.3. Molecular Features of SCLC Subtypes

Molecular and microenvironmental features of the three SCLC cohorts were evaluated using available gene signatures for individual cases with respect to the six SCLC subtypes. Features included the cellular composition of the tumor microenvironment such as immune and stromal ESTIMATE scores [30] and the abundance of stromal and immune cells estimated by MCPcounter [31]. Gene signatures were also used to estimate the level of tumor hypoxia [32] and various types of immune activity including immune cell subsets [33], immune signaling molecules [33], T-cell signature [34], tertiary lymphoid structure [35], interferon-gamma (IFN-γ) signature [36], and cytolytic signatures [37]. Signature scores were estimated by the ssGSEA algorithm as available in GSVA R packages [38]. SCLC-I and SCLC-M were also compared by GSEA using c2cp-KEGG gene sets available in the MSigDB database.

### 2.4. Deconvolution of Expression Profiles

Deconvolution of bulk-level RNA-seq data into cell type-specific gene expression patterns was performed using the CIBERSORTx algorithm [39]. We used a signature matrix representing the four cell types (the ‘TR4’ signature in the CIBERSORTx website; https://cibersortx.stanford.edu/) (accessed on 12 September 2022) of epithelial cells, fibroblasts, endothelial cells, and immune cells (represented by the cell surface markers of EPCAM, CD10, CD31, and CD45, respectively). Differential expression of genes between SCLC subtypes (i.e., SCLC-M vs. SCLC-I) was evaluated per deconvoluted cell type and was subjected to functional enrichment analysis by GSEA. SCLC-M and SCLC-I epithelial scores as mean expression levels of gene classifiers (i.e., 100 differentially expressed genes between SCLC-M and SCLC-I epithelial cells) were estimated for seven SCLC cell lines initially classified as SCLC-I tumors to determine their tumor-intrinsic cellular identity.

### 2.5. Pseudospatial Trajectory Analysis

Time-course murine SCLC single cell RNA-seq data [40] were obtained from the Gene Expression Omnibus (GEO accession of GSE149179) as feature-barcoded matrices of unique molecular identifier (UMI) counts as processed output of 10x Chromium data. Single cell RNA-seq data were obtained from cells captured at six time points (4, 7, 11, 14, 17, and 21 day) representing the NE to non-NE transition in culture with non-tumor cells depleted as described previously [40]. Unsupervised pseudotime ordering of combined tumor cells and trajectory inferences were also conducted as described previously [40]. Pseudotime trajectory analysis was also performed for human SCLC tumors using deconvoluted epithelial cell expression profiles. NE scores representing the level of transcriptional shift from NE to non-NE were estimated using genes identified previously [41]. Deconvoluted epithelial cells of human SCLC tumors and murine SCLC single cells were aligned along the inferred pseudotime and the level of expression concordance was visualized by a heatmap of pairwise correlation coefficients. Genes representing the NE-/non-NE pathway as well as those involved in EMT, Hippo/Yap1 and Notch/Rest signaling were analyzed along the inferred pseudotime.

### 2.6. Histology and Immunohistochemistry

We retrospectively reviewed data from histologically confirmed SCLC patients treated between January 2018 and August 2021 at Seoul St. Mary’s Hospital, Korea. Among the total 164 SCLC patients enrolled, 90 patients were finally included with reviewing pathology and tissue availability after approval by the Institutional Review Board of Seoul St. Mary’s Hospital and informed consent was waived due to the retrospective nature of the study (Permit number; KC21SASI0873). Consecutive 4 μm thick sections from formalin-fixed, paraffin-embedded tissue were employed for immunohistochemistry of vimentin and CD8. Vimentin and CD8+ TIL (tumor-infiltrating lymphocyte) were evaluated by immunohistochemistry using Ventana BenchMark ULTRA (Ventana Medical System, Roche, Tucson, AZ, USA). Vimentin (RTU, clone V9, Ventana Medical System) and CD8 (RTU, clone SP57, Ventana Medical System) were used as primary antibodies. The vimentin immunohistochemistry results were evaluated by examining cytoplasmic staining. H score was calculated by multiplying proportion and intensity. Proportion was evaluated as the percentage of positive tumor cells (0–100). The numbers of CD8+ TILs were counted under high magnification (×400) with five fields in the intratumoral area and the mean number of CD8+ TILs was calculated. CD8+ TIL were compared between baseline background factors using Mann–Whitney *U* tests.

## 3. Results

### 3.1. Transcriptionally Defined SCLC Subtypes

We obtained publicly available SCLC expression profiles of 81 SCLC tumors [5] to molecularly define SCLC subtypes. Expression data were subject to non-negative matrix factorization (NMF) for unsupervised discovery of SCLC subtypes [24]. To determine the number of potential molecular subtypes in the datasets, we performed subsampling-based permutation tests to obtain the distribution of stability measures (i.e., cophenetic scores) across different numbers of clusters (*k* = 2 to 10; Figure 1A) [42]. Stability scores showed a substantial drop at *k* = 4, consistent with the previous observation of four SCLC subtypes (SCLC-A, -N, -P, and -I) [14]. Notably, stability scores also decreased substantially after six clusters (*k* = 6), suggesting that additional SCLC subtypes were present in the datasets. Accordingly, we clustered SCLC transcriptomes into six subtypes (annotated as NMF1 to NMF6) and performed single sample GSEA analyses of metagenes or NMF bases as representative of subtype-specific gene expression patterns. The top three enriched molecular functions (Hallmark gene sets in MSigDB) are shown with their normalized enrichment scores in Figure 1B. NMF1 tumors were highly enriched for genes involved in angiogenesis, coagulation, and the EMT as molecular functions activated in NMF1 SCLC tumors. NMF2 tumors were enriched for immune-related functions of interferon-α- and -γ, consistent with immune-activated SCLC-I tumors. The remaining SCLC tumors (NMF3 to NMF5) showed up-regulation of genes with cell-cycle functions such as G2M checkpoint genes, suggesting high cellular turnover rates. In addition, NMF6 tumors showed relative up-regulation of hypoxia-related genes. The enrichment results for the full list of Hallmark gene sets are presented in Appendix A.

To elucidate transcriptional regulators associated with individual SCLC subtypes, we performed ARACNe [26], which infers interacting partners of transcription factors or regulators based on gene expression. The top enriched transcriptional regulators (e.g., those with enrichment of co-expressed targets) in the six NMF clusters as determined by the VIPER algorithm [27] are shown in Figure 1C (−log_10_*p* values, Fisher’s exact test). NMF1 tumors were enriched for *YAP1*- and *WWTR1*/*TAZ*-co-expressed targets, suggesting that the *YAP1*/*TAZ*-driven transcriptional network plays a regulatory role in NMF1 tumors. Thus, NMF1 SCLC tumors are driven by *YAP1*/*TAZ* and have elevated EMT activity. Previously recognized subtype-specific transcription factors such as *ASCL1* and *POU2F3* were also identified as potential regulators in NMF3 and NMF4 tumors, respectively. Expression levels of *YAP1*/*TAZ*, *ASCL1*, *POU2F3*, and *NEUROD1* are shown in Appendix A. The expression levels of cluster-specific transcriptional regulators are also shown in Appendix A. We also estimated signature-level scores of genes associated with immune function, EMT, and hypoxia in individual tumors across the six SCLC subtypes. The signature scores and levels of the four transcription regulators for the six NMF SCLC subtypes are shown in Figure 1D. NMF1 tumors were distinguished from NMF2 tumors by relatively higher EMT scores and *YAP1* levels but depleted immune signature scores. Hypoxia-representing signature scores were elevated in NMF6 tumors, albeit to a lesser extent than in NMF3 tumors. Taken together, we annotated the six NMF SCLC tumors (NMF1 to NMF6) as SCLC-M (NMF1; mesenchymal, elevated *YAP1* activity and EMT scores), SCLC-I (NMF2; inflamed, elevated immune scores), SCLC-A/-P/-N (NMF3–5; elevated activities of *ASCL1*, *POU2F3*, and *NEUROD1*, respectively), and SCLC-H (NMF6; elevated tumor hypoxia scores). Survival analysis further showed that patients with SCLC-M and SCLC-H tumors had worse clinical outcomes than patients with the other four tumor subtypes (Figure 1E). The relationships between the six NMF clusters and the previously recognized four transcriptional regulator-based SCLC subtypes (SCLC-A, -N, -P, and -Y) [7] and four NMF-based SCLC subtypes (SCLC-A, -N, -P, and -I) [14] are illustrated in Appendix A. In the comparison, four and three SCLC-M tumors (66.7% and 50.0%, respectively) were derived of SCLC-I and SCLC-N tumors according to the previous system.

To characterize the molecular and microenvironmental features of the six SCLC subtypes across cohorts, we obtained two SCLC gene expression datasets including 31 primary SCLC [7] and 50 SCLC cell lines [43]. Using the expression profiles obtained for the six SCLC subtypes from the first dataset (‘George et al.’) as the training set, the six SCLC subtypes were annotated for two additional datasets (‘Rudin et al.’ and ‘CCLE’, respectively) using PAM [28]. The landscape of molecular and cellular features with respect to cluster membership across the three datasets is shown in Figure 2. The prevalence of each of the six SCLC subtypes in the two primary SCLC tumor datasets was 3.1–7.4% (SCLC-M, from ‘George’ and ‘Rudin’ datasets, respectively), 18.5–25.0% (SCLC-I), 24.7–31.3% (SCLC-A), 16.0–18.8% (SCLC-P), 21.9–27.2% (SCLC-N), and 0–6.2% (SCLC-H), suggesting that SCLC-M tumors are a minority subtype compared with SCLC-I tumors. Of note, examination of cell lines did not support the presence of SCLC-I tumors in contrast to the enrichment of SCLC-M cell lines (14.0% and 0% for SCLC-M and SCLC-I tumors in the ‘CCLE’ dataset, respectively). This suggests that SCLC-I tumors are largely determined by the presence of tumor-infiltrating lymphocytes in the tumor microenvironment and it is possible that SCLC-I tumors deprived of stromal or immune cells may resemble SCLC-M tumors.

To profile the TME architecture with respect to SCLC subtype, we used ESTIMATE [30] and MCPcounter [31] to estimate the level of immune and stromal cells in individual tumor specimens. Overall, SCLC-I types showed relative enrichment of immune-related signals (e.g., immune scores/ESTIMATE and immune cells/MCPcounter) compared with SCLC-M tumors. However, transcriptional signals from stromal cells (e.g., stromal scores/ESTIMATE and fibroblasts/MCPcounter) were comparable between SCLC-M and SCLC-I tumors. This trend was consistent among the two primary SCLC RNA-seq datasets (‘George’ and ‘Rudin’) for individual marker genes and signature scores representing immune- and EMT-related signaling pathways. For example, *IFNG* and genes encoding cytolytic enzymes (*GZMA*, *GZMB* and *PRF1*) were relatively up-regulated in SCLC-I tumors, indicating that SCLC-I tumors are characterized by a high level of tumor-infiltrating lymphocytes with potential cytolytic activity. With regard to EMT markers, we observed that *VIM* and *AXL* genes were commonly expressed in both SCLC-M and SCLC-I tumors but that canonical EMT-related transcription factors such as *ZEB1*, *ZEB2*, *TWIST* and *SNAIL* were relatively up-regulated only in SCLC-M tumors. Signatures representing various types of antitumor immunity including the activity of immune cell subsets [33], T-cell signature [33], and tertiary lymphoid structure [35] were relatively higher in SCLC-I tumors than in SCLC-M tumors. We also observed that previously proposed NE and non-NE markers (e.g., *CHGA*, *SYP* and *REST*) as well as SCLC subtype-specific markers (e.g., *MYC*, *BCL2*, *DLL3*, and *TFF1*) showed variable expression across SCLC subtypes, suggesting that these are not suitable for distinguishing between SCLC-M and SCLC-I tumors as single markers. Mutation profiles of SCLCs with respect to tumor subtype are shown in Appendix A. Somatic mutations were largely concordant across SCLC subtypes with universal *TP53* and *RB1* mutations. GSEA of additional gene sets (KEGG-c2cp, MSigDB) further highlighted extracellular matrix (ECM) as one of the most distinguishing functions of SCLC-M tumors when compared with SCLC-I tumors (*p* = 0.033, Kolmogorov–Smirnov test; Appendix A). Among the genes belonging to ECM gene sets, *ITGB5*, *CD36* and *LAMA1* were relatively up-regulated in SCLC-M tumors.

### 3.2. Tumor-Intrinsic Transcriptional Differences between SCLC-M and SCLC-I Tumors

The absence of SCLC-I tumors in the cell line datasets may be largely due to the lack of stromal and immune cells in these cultures. This suggests that a classifier built on primary tumors including the tumor microenvironment may not capture the intrinsic tumor-specific features of SCLC-M and SCLC-I tumors. To further identify tumor-intrinsic, subtype-specific features, we employed deconvolution-based imputation of cell type-specific gene expression using CIBERSORTx [39]. Expression profiles of four cell types were imputed from individual bulk-level RNA-seq data representing epithelial cells (EPCAM), fibroblasts (CD10), endothelial cells (CD31), and immune cells (CD45). These results are presented in Appendix A. First, we performed GSEA analysis for deconvoluted epithelial cells in SCLC-M and SCLC-I tumors to identify tumor-intrinsic differences between these two SCLC subtypes. The top five enriched molecular functions in SCLC-M and SCLC-I tumors are shown in Figure 3A. As expected, genes associated with the EMT, angiogenesis, and TGFβ signaling pathways were relatively up-regulated in SCLC-M tumor epithelial cells compared with SCLC-I tumor epithelial cells. Of note, the deconvoluted tumor epithelial cells of SCLC-I tumors showed up-regulation of genes with immune-related functions such as allograft rejection and the interferon-gamma response. Enrichment plots of the representative functions of SCLC-M (‘EMT’) and SCLC-I tumors (‘allograft rejection’) are shown in Figure 3B. These findings suggest that immune-related functions are also up-regulated in SCLC-I tumor epithelial cells as tumor-intrinsic features in addition to the infiltration of immune cells in the TME of SCLC-I tumors. GSEA results for epithelial cells, fibroblasts, endothelial cells, and immune cells in SCLC-M tumors compared with those of SCLC-I tumors are presented in Appendix A. We observed that fibroblasts and endothelial cells of SCLC-M may be dysfunctional in terms of oxidative phosphorylation and the cell cycle, respectively, compared with fibroblasts and endothelial cells of SCLC-I tumors.

The gene expression heatmap of genes differentially expressed between SCLC-M and SCLC-I epithelial cells is shown in Appendix A. Selected genes were further used to derive SCLC-M and SCLC-I tumor-intrinsic scores (designated as M- and I-scores, respectively) for seven SCLC-M cell lines (Figure 3C). M and I scores of the seven SCLC-M cell lines were inversely correlated (Pearson correlation coefficient of −0.58136, *p* value = 0.00001) and the cell lines were further sorted according to M- vs. I- differential scores. These findings further suggest that SCLC-M cell lines are not exclusive to SCLC-I tumors, some of which may harbor tumor-intrinsic features SCLC-I tumors. Figure 3D shows candidate genes whose expression was correlated with the M- vs. I- differential scores both in SCLC tumor (SCLC-M and SCLC-I) epithelial cells and SCLC-M cell lines. Pearson correlation coefficients with significance levels for selected genes are shown in Appendix A. For example, *ITGB1* (*CD29*) encoding integrin subunit beta 1 was transcriptionally up-regulated in SCLC-M primary epithelial cells and SCLC-I cell lines with high M scores. Integrin-mediated signaling is required for the TGFβ-promoted EMT progression [44] along with the direct impact of ITGB1 on known markers of EMT [45]. *SH2B2* was up-regulated in SCLC-I primary epithelial cells and SCLC-I cell lines with high-level I scores. *SH2B2* has previously been recognized as a negative regulator of cytokine signaling [46], thus its transcriptional up-regulation in epithelial cells and cell lines indicates that it may be a marker for SCLC regardless of the composition of the tumor microenvironment.

Next, to validate the presence of SCLC-M subtype in SCLC tumors, we evaluate the expression of vimentin and the CD8+ TIL score in the SCLC tumor samples. Clinical characteristics were summarized in Appendix A. Based on tumor-intrinsic features and characteristics of microenvironment in SCLC, we analyzed the CD8+ TIL score and tumor vimentin expression. The CD8+ TIL score was a representative marker of the immune phenotype in many solid tumors including SCLC [14]. Vimentin was a representative marker of the EMT phenotype; the classic type SCLC did not express vimentin [47]. Any positivity of the vimentin stain was considered as vimentin expression. A count of 12/90 (13.3%) tumors expressed vimentin (Appendix A). The mean CD8+ TIL score was significantly high in vimentin-expressed tumors (27.5 vs. 12.0, *p* = 0.0154, Figure 4A). However, among the vimentin-expressed tumors, the CD8+ TIL score was dichotomized into a high and low group (Figure 4B). The CD8+ TIL low/Vimentin(+) tumors (6 out of 90 SCLC patients corresponding to 6.7% of the cohort) can be classified as SCLC-M with an elevated EMT signature but lack of immune cell infiltration, compared with SCLC-I with elevated immune cell infiltration.

### 3.3. Cross-Species Analysis of SCLC Progression

Single cell sequencing data of murine SCLC cell lines representing the MYC-driven early-to-invasive progression at distinct time points (4 to 21 days of culture) were obtained to enumerate the cellular NE-to-non-NE transition of SCLC tumors [40]. The linear trajectory of murine SCLC cells was inferred along with the pseudotime of cells as reported previously (Figure 5A). A clear distinction between early-passage cells (4 and 7 days) and late-passage cells (11 to 21 days) was observed along the inferred pseudotime (Figure 5B). Of note, we also observed a linear trajectory for human SCLC tumor epithelial cells encompassing the transition of NE-to-non-NE SCLC tumors (Figure 5C). The occurrence of SCLC subtypes in order of pseudotime highlights the fact that NE-tumors (SCLC-A and SCLC-N) and non-NE-tumors (SCLC-M and SCLC-P) are frequent during early and late development, respectively, whereas the remaining SCLC subtypes of SCLC-I and SCLC-H tumors appear to be intermediate tumor types (Figure 5D). The transcription-level concordance heatmap shows the cross-species correlation between the expression patterns of mouse cells (*y*-axis; 19,366 cells ordered by pseudotime) and those of human SCLC tumors (*x*-axis, 81 tumors ordered by pseudotime) (Figure 5E). NE scores of individual cells or tumors were calculated as described previously [40] and are shown according to pseudotime. For both species, the inferred pseudotime was in overall agreement with the changes in estimated NE scores indicating that trajectories largely reflect the NE-to-non-NE transition. The marked similarity between these mouse and human tumors in terms of inferred pseudotime suggests that human SCLC development representing the NE-to-non-NE transition recapitulates the mouse SCLC tumor early-to-invasive tumor progression. Expression levels of selected NE markers as well as those involved in the EMT, Hippo/YAP1, and Notch/REST signaling are shown for human SCLC tumors in order of pseudotime (Figure 5F). While NE markers showed decreasing patterns along pseudotime consistent with the NE-to-non-NE transition, EMT markers as well as those in Hippo/YAP1 and Notch/REST signaling pathways showed pseudotime-dependent up-regulation. This suggests that the NE-to-non-NE transition in murine SCLC tumors recapitulates the developmental trajectory of human SCLC tumors and allows discrimination of SCLC subtypes in terms of developmental order as represented by pseudotime.

## 4. Discussion

During the EMT, tumor epithelial cells acquire motility and the ability to invade adjacent tissues, which are fundamental to the metastatic dissemination of tumor cells [48]. Thus, activation of the EMT is largely considered to indicate a poor prognosis and ‘mesenchymal’ tumor subtypes with elevated EMT activity have been observed across various tumor types [15,16]. EMT has been proposed as a potential mechanism underlying the therapeutic resistance of SCLC tumors [17,49]; however, the roles of EMT in SCLC metastasis as well as the prevalence of EMT activation in SCLC tumors are unclear. Moreover, some previous studies have proposed that the EMT activity of SCLC tumors is associated with favorable clinical outcomes and is compatible with antitumor immune activity. For example, SCLC-I tumors showed elevated EMT activity with low and high levels of *CDH1* and *VIM* as epithelial and mesenchymal markers, respectively [14]. Although the relative stromal score is both enriched in SCLC-I and SCLC-M tumors, the latter was functionally correlated with EMT, angiogenesis, and coagulation suggestive of SCLC-M tumor-specific functional characteristics. In addition, *YAP1* transcriptional regulators are known to upregulate the EMT activity of lung cancers [50], but *YAP1* expression is enriched in limited-stage tumors with an inflamed phenotype [51]. These discrepancies in the role of EMT activity in SCLC tumors may be due to the lack of resolution of the current SCLC subtyping system. Although SCLC-I tumors were recognized as ‘triple-negative’ SCLCs without *ASCL1*, *NEUROD1*, and *POU2F3* expression, these tumors have also been proposed to be additional subtypes that express different transcription factors such as *ATOH1* [52]. In this study, we employed an NMF algorithm to refine the molecular taxonomy of SCLC tumors and identified new SCLC-M subtypes that are distinct from previously proposed SCLC-I tumors in terms of various molecular and clinical features. However, our study demonstrates that SCLC-M tumors are deprived of antitumor immunity and are also associated with unfavorable clinical outcomes, suggesting that triple-negative SCLC tumors may be heterogeneous and considered for different therapeutic strategies.

It is unclear whether the transcriptional signal from so-called mesenchymal tumors originates from the tumor itself or the microenvironment as tumor-intrinsic or -extrinsic features, respectively. Early reports of mesenchymal tumors of brain and ovarian tumors proposed that mesenchymal tumors are characterized by tumor-intrinsic EMT activity with a predisposition to metastases and unfavorable clinical outcomes [53,54]. However, this has been challenged by studies reporting that tumor-infiltrating stromal cells are responsible for the transcriptional signals of mesenchymal tumors in colorectal cancers [16]. In the current study, signature-based analyses of SCLC-M transcriptomes suggested that these tumors have a high level of EMT activity as well as tumor-infiltrating fibroblasts as inferred by cell type-specific signatures (Figure 2). However, deconvolution analyses of SCLC tumors further indicated that tumor-intrinsic EMT activity is up-regulated in SCLC-M tumors, suggesting that SCLC-M tumors are distinguished by tumor-intrinsic EMT features instead of their fibroblast content. Although SCLC-I tumors are largely determined by the elevated level of tumor-infiltrating immune cells, those transcriptional signals will not be identified in cell lines deprived of tumor microenvironments. However, we observed that SCLC-I cell lines show the enrichment of immune-related gene sets (e.g., allograft rejection and interferon signalings) suggesting that immune-enriched or inflamed tumor cells also show distinctive immunological features. However, this requires further validation in platforms other than bulk-level transcriptomes such as patient-derived xenografts or single tumor cells amenable to the deconvolution of cell type-specific expression.

Trajectory analyses showed that the pseudotime inferred from the human SCLC tumor epithelial cells was largely concordant with that of an experimental NE-to-non-NE murine SCLC model. In addition to the developmental relationship among known SCLC subtypes of SCLC-A, -N (both as early NE tumors) and -P (late non-NE tumors), the analyses suggested that SCLC-I tumors are developmental intermediates while SCLC-M tumors are more similar to SCLC-P, non-NE tumors. This developmental relationship provides insight into the molecular features of our two newly identified SCLC subtypes. For example, it is possible that SCLC-I tumors are tumors in the NE-to-non-NE transition with a modest level of EMT activity in addition to infiltrating immune cells. SCLC-M tumors acquire a substantial level of EMT activity during later stages of the NE-to-non-NE transition, but this transition is also accompanied by the depletion of immune cells, most likely due to altered cellular interactions in the tumor microenvironment. This hypothesis is consistent with a report where a high level of immune infiltration induces EMT followed by up-regulation of immune checkpoints such as *CTLA-4* and *PD-L1* in lung adenocarcinomas [55]. The immune–EMT axis may represent a complicated cellular crosstalk underlying cancer progression [56], and further investigation is required to elucidate the signaling cascade in the SCLC-M and SCLC-I tumors.

The enriched molecular functions of NMF3–NMF5 clusters are similar and they share a common function of “G2M checkpoint” suggesting that tumor cells of NMF3–NMF5 clusters will have increased cellular proliferation with shortened cell doubling time instead of representing the cluster-specific functions. Molecular studies using human SCLC (hSCLC) cell lines showed that *ASCL1* and *NEUROD1* are required for the survival of the cells’ increased tumor-initiating capacity in xenograft assays suggesting that these factors lead to similar cellular consequences [57,58,59]. Thus, the molecularly distinguishing features of these clusters can be identified by searching for transcriptional regulators such as *ASCL1, NEUROD1*, and *POU2F3* while these transcriptional regulators share common cellular consequences.

It has been reported that *VIM* and *AXL* were relatively up-regulated in triple transcription factor-negative SCLC (also defined as SCLC-I by Gay et al. [14]). In addition, it was reported that classic SCLC cell lines do not express *VIM* while variant SCLC cell lines do express *VIM* [47]. Given the relative paucity of SCLC clinical specimens eligible for genetic testing, *VIM* genes being well-recognized EMT effectors are preferred as single markers evaluating the level of EMT. In this study, we demonstrated that *VIM* staining can be jointly performed with CD8+ TIL quantification so that SCLC-M tumors with Vim+ and a low CD8+ TIL score can be identified in clinical cohorts.

In addition, coexpression-based network analyses have been employed to elucidate the roles of subtype-specific transcriptional regulators as master regulators. Known transcriptional regulators including *ASCL1* and *POU2F3* were identified along with *YAP1*/*WWTR1* for SCLC-M subtypes. Of note, *NEUROD1*, the SCLC-N-specific transcriptional regulator, was not identified in the top significant list. This may be due to the less apparent transcriptional regulation of *NEUROD1* in NMF5 tumors, but can also be dataset- or method-related issues such as biases due to the ascertainment and indirect interactions, respectively. Thus, it will require further investigation whether the transcriptional up-regulation of transcriptional regulators of SCLC subtypes represents the roles of master regulators.

Previous pan-cancer scaled analyses revealed that tumor hypoxia is prevalent and heterogeneous across tumor types including lung cancers [60]. We also identified novel SCLC-H subtypes in this study potentially representing a subset of SCLC with elevated levels of tumor hypoxia. The annotation is based on signature scores representing cellular hypoxia. Tumor hypoxia is associated with resistance to cancer therapy, likely explaining the worse clinical outcomes of SCLC-H tumors [61]. However, the absence of SCLC-H tumors in the independent datasets raises concerns regarding the robustness of this tumor designation. However, it is also possible that the level of tumor hypoxia reflected by hypoxia signature scores is not able to define the distinctive tumor subsets given that hypoxia signature scores are variable across the SCLC subtypes.

## 5. Conclusions

Our study demonstrates that triple-negative SCLC tumors (e.g., no expression of *ASCL1*, *NEUROD1*, and *POU2F3*) are heterogeneous and can be subclassified into SCLC-I and SCLC-M tumors. Although these two molecular subtypes may be evolutionarily related, the molecular and clinical properties of them require clinical attention in evaluating the SCLC patients.

## Figures and Tables

**Figure 1 cancers-14-05600-f001:**
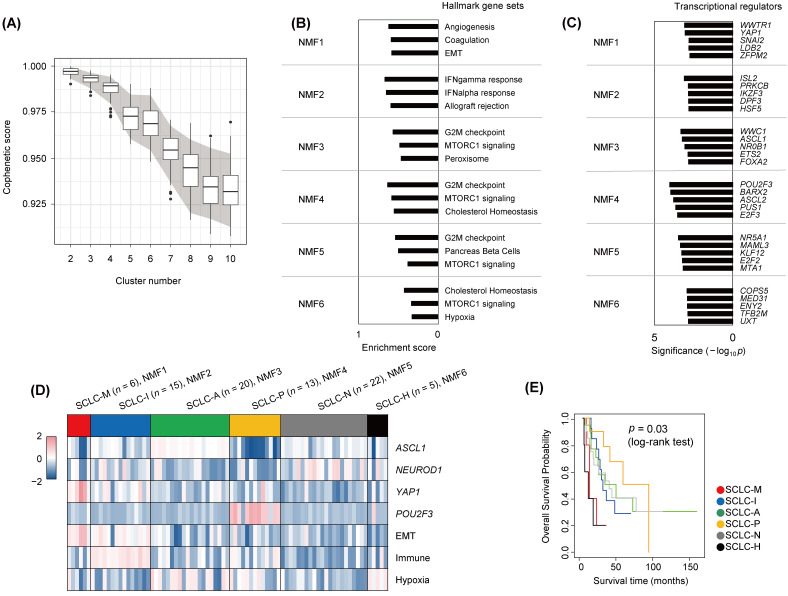
Transcriptional analysis to define SCLC subtypes. (**A**) Boxplots showing ranges of cophenetic correlation coefficients (*y*-axis) measured across *K* values (2 to 10; *x*-axis). Gray area indicates 90 percentiles (5% and 95%) of cophenetic scores per *K* value as obtained from permutation tests. (**B**) For six NMF clusters (NMF1 to NMF6), the top three enriched molecular functions (Hallmark gene sets in MSigDB) are shown with enrichment scores. (**C**) Cluster-specific transcriptional regulators are shown per NMF cluster. The level of significance was estimated by the VIPER algorithm and is shown on the *x*-axis (−log_10_*p*). (**D**) Six NMF clusters (NMF1 to NMF6) are annotated SCLC-M, -I, -A, -P, -N, and -H, respectively, according to the level of expression of four transcriptional regulators (*ASCL1*, *NEUROD1*, *YAP1*, and *POU2F3*) and the signature scores of three molecular functions (EMT, immune activity, and hypoxia). (**E**) Kaplan–Meier survival curves of overall survival are shown for the six SCLC subtypes. Significance was estimated by log-rank test.

**Figure 2 cancers-14-05600-f002:**
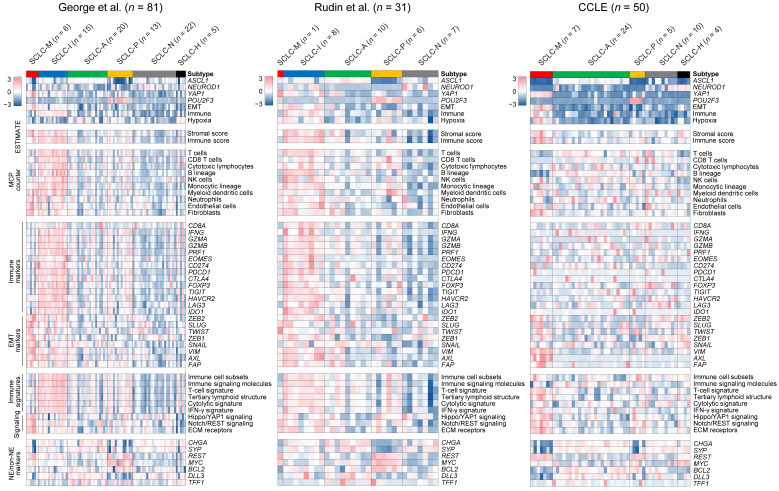
Molecular features of SCLC subtypes. Various molecular and cellular features of the six SCLC subtypes are shown as heatmaps. Along with the primary dataset used for subtype discovery (George et al., **left**), results for two additional cohorts (Rudin et al. (**middle**) and CCLE (**right**)) are presented. Levels of the four transcriptional regulators and three molecular functions indicated in Figure 1D are shown. Stromal-immune scores (‘ESTIMATE’) and the abundance of 10 immune and stromal cell types (‘MCPcounter’) are also shown. Relative immune activities are shown for 14 immune genes (‘Immune markers’) and six immune pathways (‘Immune signatures’). In addition, expression levels of eight EMT-related (‘EMT markers’), seven genes associated with SCLC tumorigenesis (‘NE/non-NE markers’), and three molecular functions (‘Signaling’) are shown.

**Figure 3 cancers-14-05600-f003:**
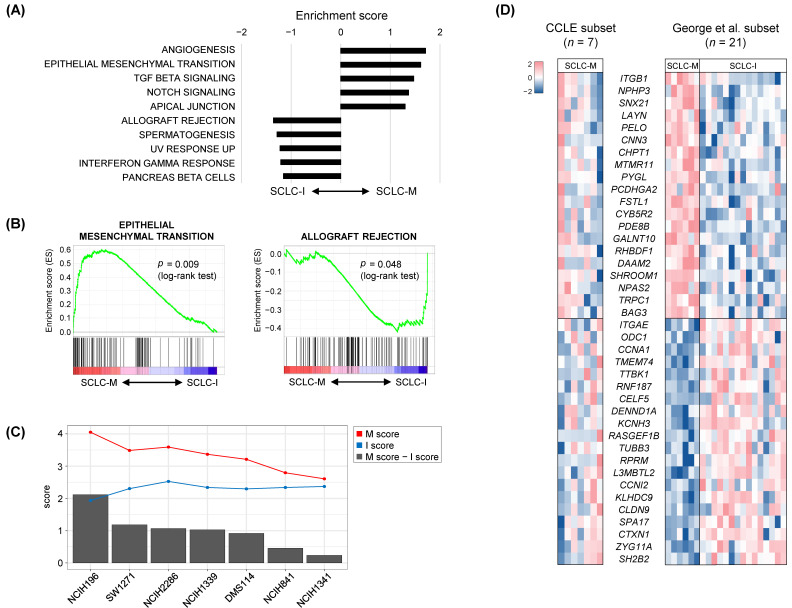
Tumor-intrinsic features of SCLC-M and SCLC-I tumors. (**A**) Ten molecular features are shown as those either up-regulated or down-regulated in deconvoluted SCLC tumor epithelial cells of SCLC-M compared with those of SCLC-I. (**B**) Enrichment plots of the selected functions of SCLC-M and SCLC-I are shown. (**C**) For the seven SCLC-M cell lines, tumor-intrinsic M and I scores are shown (red and blue lines, respectively). Bar plots indicate differential M and I scores. (**D**) The heatmap shows selected genes highly correlated with the differential M and I scores for cell lines and primary SCLC tumors (**left** and **right**, respectively).

**Figure 4 cancers-14-05600-f004:**
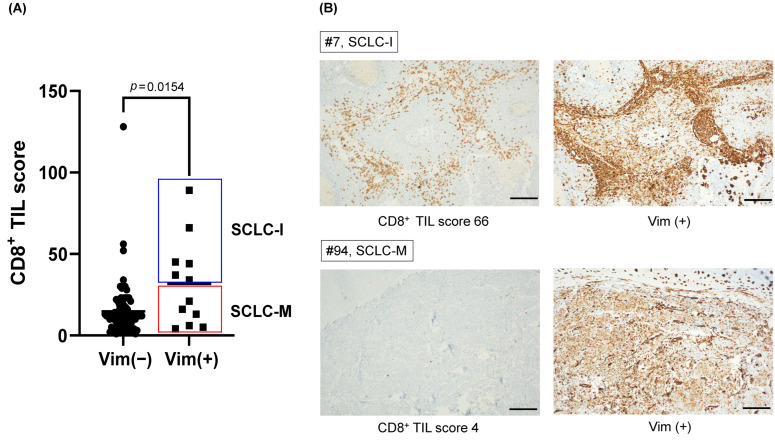
The prevalence of SCLC-M subtype in primary SCLC tumor cohort. (**A**) Mean CD8+ TIL scores of vimentin (+) tumors and vimentin (−) tumors. CD8+ TIL score of vimentin (+) tumors divided into SCLC-I and SCLC-M subtypes. (**B**) Representative immunohistochemical stains of CD8+ TIL high/Vimentin (+) (SCLC-I) and CD8+ TIL low/Vimentin (+) (SCLC-M) tumors. (scale bar; 100 μm).

**Figure 5 cancers-14-05600-f005:**
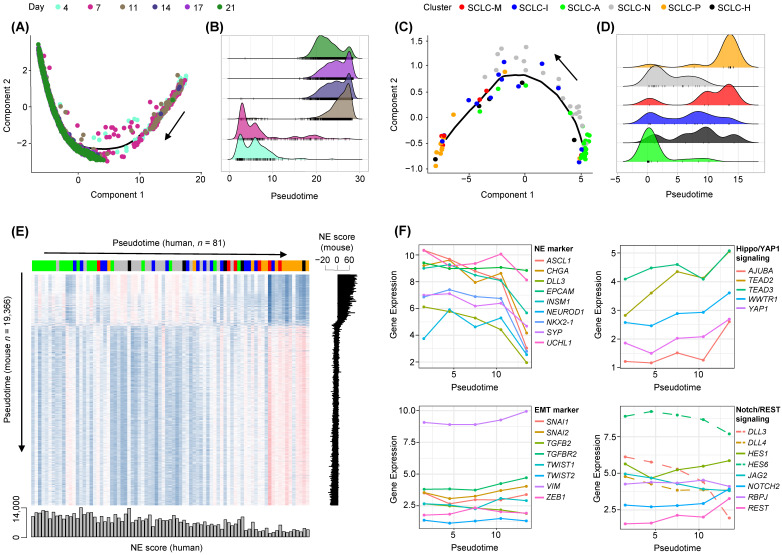
Cross-species analysis of NE to non-NE progression of human and mouse SCLC tumors. (**A**) Pseudotime trajectory of murine SCLC single cell RNA-seq data shows the ordering of cells from early to late time points (Day 4 to Day 21). Black arrow shows an increase in pseudotime. (**B**) Cell abundance is plotted according to pseudotime and segregated cells harvested early (Days 4–7) and late (Days 11–21). (**C**,**D**) Similarly, the trajectory of human SCLC tumor epithelial cells and cellular abundance along the pseudotime are plotted. Black arrow shows an increase in pseudotime. (**E**) A heatmap shows the correlation in gene expression between human SCLC epithelial cells (*x*-axis; 81 samples) and mouse single cells (*y*-axis, 19,366 cells), both sorted in order of pseudotime. NE scores representing the transcriptional shift from NE to non-NE decreased consistently with pseudotime. (**F**) Levels of expression (*y*-axis, log2-scaled) are shown for NE and EMT markers as well as those belonging to Hippo/YAP1 and Notch/REST signaling pathways. Dashed lines in the Notch signaling pathway indicate Notch-inhibitory genes.

## Data Availability

SCLC RNA-seq data in this study were obtained from the European Genome-phenome Archive (EGAS00001000925 and EGAS0000100488). SCLC cell line data are available from the Cancer Cell Line Encyclopedia (CCLE) database (https://sites.broadinstitute.org/ccle/datasets) (accessed on 8 September 2022) [21]. Murine SCLC single cell RNA-seq data are available in the GEO database (GSE149179).

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
