# Peer review of "Transcriptional Profiling Reveals Mesenchymal Subtypes of Small Cell Lung Cancer with Activation of the Epithelial-to-Mesenchymal Transition and Worse Clinical Outcomes"

_cancers, 2022, doi:10.3390/cancers14225600_

Round 1

Reviewer 1 Report

The authors used online gene expression profiles of SCLC primary tumors to redo the molecular subtyping of SCLC tumors and characterized two novel subtypes including SCLC-M tumors with elevated epithelial-to-mesenchymal transformation (EMT) and YAP1 activity but low level of anti-cancer immune activity and SCLC-H tumors with high levels of hypoxia. Then they strengthened that newly identified SCLC-M tumors are biologically and clinically distinct from SCLC-I tumors which should be taken into accounts for the diagnosis and treatment of the disease. The whole study design and results shown in this article is interesting and fantastic, however, several major concerns should be addressed.

1.     Several typo errors exist in the manuscript such as ‘discovery’ in the first sentence of the summary.

2.     From Line 198 to 201, the authors explained why k=6 was chosen however the stability scores were decreasing rapidly from 4 to 8, but kept stable after 9. So why was k was defined as 6 but not 9?

3.     Top enriched transcriptional factors in each cluster were shown in Fig 1c, however, NEUROD1 was not listed as the top factor in NMF5, so whether NMF5 could be defined as SCLC-N should be questioned in this dataset.

4.     Additionally, I suggest that the expression levels of those top enriched TFs should be compared among those clusters.

5.     Since NMF1 was highly correlated with EMT, angiogenesis, and coagulation, the relative stromal scores are suggested to be shown among subclusters although they are consistent between NMF1 and NMF2.

6.     Additionally, the extracellular matrix signals including integrin family members or the potential roles of cancer associated fibroblasts are suggested to be closely looked at.

7.     The authors chose several key TFs or representative markers for the SCLC-M and SCLC-I tumors, however they validated the existence of SCLC-M using vimentin only, which is inappropriate.

8.     Whether the novel SCLC-M subtype could be potentially challenged with licensed drugs is suggested to be evaluated.

Reviewer 2 Report

Cho et. al. in their manuscript titled “Transcriptional profiling reveals mesenchymal subtypes of small cell lung cancer with activation of epithelial-to-mesenchymal transition and worse clinical outcomes”, describe an in silico approach by which they identify multiple subtypes of small cell lung cancer (SCLC). They used published RNA-seq datasets from two different patient cohorts and the cancer cell line encyclopedia (CCLE). Using NMF-based unsupervised subtyping analysis, they discovered six SCLC subtypes each with unique gene expression patterns. They focused on the mesenchymal (M) and the inflamed (I) subtypes which have a lot of shared characteristics but differ based on the tumor extrinsic immune signature supplied by the infiltrating immune cells. Finally, using a pseudotime-based analysis of murine SCLC cells they identify the M subtype to have progressed to a more aggressive non-neuroendocrine nature from neuroendocrine subtypes.

Overall, the study elucidates hitherto unknown SCLC phenotype M with epithelial-to-mesenchymal (EMT) features that are distinct from immune infiltrated I subtype which was previously described. This M subtype has the worst prognosis and might benefit from more targeted therapies and better management.

There are a few minor comments that will make the manuscript better:

1)    The authors need to discuss the NMF3-5 clusters as the pathway analysis show they affect similar ones.

2)    Change the nomenclature of the newly identified subtypes as it is confusing to tease apart the new from the old.

3)    The authors need to verify the existence of the H subtype in another SCLC cohort as it was not detected in the test (Rudin) cohort.

4)    In the deconvolution analysis of the SCLC-I tumors, spermatogenesis and pancreas beta cells pathways were highly upregulated. The authors should comment on the significance of these outcome in the results or discussion.

5)    Is the I subtype a subset of the M subtype? It seems so from the IHC Vimentin and CD8 staining results as shown in Figure 4. The authors should discuss this in more detailed manner.

Author Response

Cho et. al. in their manuscript titled “Transcriptional profiling reveals mesenchymal subtypes of small cell lung cancer with activation of epithelial-to-mesenchymal transition and worse clinical outcomes”, describe an in silico approach by which they identify multiple subtypes of small cell lung cancer (SCLC). They used published RNA-seq datasets from two different patient cohorts and the cancer cell line encyclopedia (CCLE). Using NMF-based unsupervised subtyping analysis, they discovered six SCLC subtypes each with unique gene expression patterns. They focused on the mesenchymal (M) and the inflamed (I) subtypes which have a lot of shared characteristics but differ based on the tumor extrinsic immune signature supplied by the infiltrating immune cells. Finally, using a pseudotime-based analysis of murine SCLC cells they identify the M subtype to have progressed to a more aggressive non-neuroendocrine nature from neuroendocrine subtypes.

Overall, the study elucidates hitherto unknown SCLC phenotype M with epithelial-to-mesenchymal (EMT) features that are distinct from immune infiltrated I subtype which was previously described. This M subtype has the worst prognosis and might benefit from more targeted therapies and better management.

There are a few minor comments that will make the manuscript better:

Point 1: The authors need to discuss the NMF3-5 clusters as the pathway analysis show they affect similar ones.

Response 1: We appreciate this comment. As commented, the enriched molecular functions of NMF3-5 clusters are similar and they share a common signature of “G2M checkpoint” suggesting that tumor cells in NMF3-NMF5 will have increased cellular proliferation with shortened cell doubling time. Thus, the molecularly distinguishing features of these clusters should be identified in a more upstream level such as transcriptional regulators such as ASCL1, NEUROD1, and POU2F3. We address this concern in the revised manuscript (pp.23, ll.555-564). also citing a previous report [57-59].

Point 2: Change the nomenclature of the newly identified subtypes as it is confusing to tease apart the new from the old.

Response 2: We appreciate this comment and agree with this concern. The aim of this study is to distinguish SCLC-M from SCLC-I tumors. In spite of the unsupervised nature of NMF clustering, Supplementary Fig. 3 shows that the cluster membership of previous SCLC subtypes such as SCLC-A, SCLC-N, and SCLC-P are largely maintained across the nomenclature system. Thus, we maintain our previous annotation system.

Point 3: The authors need to verify the existence of the H subtype in another SCLC cohort as it was not detected in the test (Rudin) cohort.

Response 3: This comment is very helpful and we agree with this concern. Compared to SCLC-M tumors, SCLC-H tumors are inconsistent with respect to cohorts and the signature scores of tumor hypoxia are also variable. As proposed, no SCLC-H tumors were found in the Rudin cohort. It is possible that the Rudin cohort is much smaller (31 patients) than those of George and thus, low-frequent SCLC-H tumors are absent in this cohort. But, it is also possible that hypoxia signatures cannot define distinctive SCLC subsets, which explains the variable and inconsistent hypoxia scores across SCLC subtypes. We address this concern in the revised manuscript (pp.24, ll.590-592)

Point 4: In the deconvolution analysis of the SCLC-I tumors, spermatogenesis and pancreas beta cells pathways were highly upregulated. The authors should comment on the significance of these outcome in the results or discussion.

Response 4: We appreciate this comment. The GSEA results of primary tumors and cell lines are different since cell lines are lacking tumor microenvironments. This requires caution, especially for SCLC-I tumors where transcriptional signatures of hallmarks of SCLC-I tumors (immune signatures) cannot be detected in cell lines. Of interest, the immune-related signatures are also observed in SCLC-I cell lines such as allograft rejection and interferon signalings, suggesting that the SCLC-I tumor cells also show immunologically distinct features compared to those of other SCLC subtypes This concern is already been discussed in our initial version of the manuscript (pp.23, ll.530-535).

Point 5: Is the I subtype a subset of the M subtype? It seems so from the IHC Vimentin and CD8 staining results as shown in Figure 4. The authors should discuss this in more detailed manner.

Response 5: We appreciate this comment. As proposed, SCLC-M tumors are suggested as a subset of SCLC-I tumors. We revised the Discussion of the manuscript with a detailed description of the results shown in Figure 4 (pp.24, ll.565-572).

Reviewer 3 Report

Manuscript ID: cancers-1951459

Type of manuscript: article

Title:  Transcriptional profiling reveals mesenchymal subtypes of small cell
lung cancer with activation of the epithelial-to-mesenchymal transition and
worse clinical outcomes

Journal: Cancer

In this article, the authors have investigated and characterize SCLC-M (mesenchymal) tumors distinct from SCLC-I tumors for molecular features, clinical outcomes, and cross-species developmental trajectories using gene expression profiles of SCLC primary tumors and cell lines. They have demonstrated that SCLC-M tumors show elevated EMT and YAP1 activity but low level of anti-cancer immune activity .

The paper is well done and presented. All the criteria for a correct presentation were respected. It is well described, and the results are conclusive and helpful.

I personally recommend the publication of this paper.

I have only a suggestion:

I suggest to describe better YAP1 for instance in the introduction,, being  that this gene is a downstream nuclear effector of the Hippo signaling pathway.

Round 2

Reviewer 1 Report

The authors have sufficiently elucidated my concerns and provided adequate supplementary data, so I suggest the final decision is acceptance.

Reviewer 2 Report

My comments have been addressed sufficiently in the revised manuscript.